# Persistent symptoms after COVID-19 are not associated with differential SARS-CoV-2 antibody or T cell immunity

Daniel M. Altmann[1] ✉, Catherine J. Reynolds[2], George Joy[3,4], Ashley D. Otter[5], Joseph M. Gibbons[6], Corinna Pade[6], Leo Swadling[7], Mala K. Maini[7], Tim Brooks[5], Amanda Semper[5], Áine McKnight[6], Mahdad Noursadeghi[7], Charlotte Manisty[3,4], Thomas A. Treibel[3,4], James C. Moon[3,4], COVIDsortium investigators* & Rosemary J. Boyton[2,8] ✉

Among the unknowns in decoding the pathogenesis of SARS-CoV-2 persistent symptoms in Long Covid is whether there is a contributory role of abnormal immunity during acute infection. It has been proposed that Long Covid is a consequence of either an excessive or inadequate initial immune response. Here, we analyze SARS-CoV-2 humoral and cellular immunity in 86 healthcare workers with laboratory confirmed mild or asymptomatic SARS-CoV-2 infection during the first wave. Symptom questionnaires allow stratification into those with persistent symptoms and those without for comparison. During the period up to 18-weeks post-infection, we observe no difference in antibody responses to spike RBD or nucleoprotein, virus neutralization, or T cell responses. Also, there is no difference in the profile of antibody waning. Analysis at 1-year, after two vaccine doses, comparing those with persistent symptoms to those without, again shows similar SARS-CoV-2 immunity. Thus, quantitative differences in these measured parameters of SARS-CoV-2 adaptive immunity following mild or asymptomatic acute infection are unlikely to have contributed to Long Covid causality. ClinicalTrials.gov (NCT04318314).

There is a substantial ongoing problem posed by the accumulating global disease burden of those suffering from persistent symptoms after SARS-CoV-2 acute infection. Long Covid describes the persistence of symptoms more than 4-weeks after acute infection with around one-fifth of cases comprising those now symptomatic beyond 2-years[1]. A number of recent studies have sought to clarify the range of symptoms, the combinations in which they occur and their progression through what is often a relapsing and remitting timecourse[2–4]. Studies identify over 200 associated symptoms, though with a core, diagnostic set encompassing fatigue/exhaustion, pain, post-exertional malaise, cardiovascular, respiratory, neurological and cognitive function.

Even documenting the Long Covid disease burden can be difficult considering that many sufferers did not access testing to confirm the initiating SARS-CoV-2 acute infection, and that there are no agreed diagnostic tests or clinical criteria to define the persistent condition. Most now estimate that Long Covid ensues from around 10% of all infections, though the incidence may be somewhat lower in

[1]Department of Immunology and Inflammation, Imperial College London, London, UK. [2]Department of Infectious Disease, Imperial College London, London, UK. [3]St Bartholomew's Hospital, Barts Health NHS Trust, London, UK. [4]Institute of Cardiovascular Science, University College London, London, UK. [5]UK Health Security Agency, Porton Down, UK. [6]Blizard Institute, Barts and the London School of Medicine and Dentistry, Queen Mary University of London, London, UK. [7]Division of Infection and Immunity, University College London, London, UK. [8]Lung Division, Royal Brompton and Harefield Hospitals, Guy's and St Thomas' NHS Foundation Trust, London, UK. *A list of authors and their affiliations appear at the end of the paper. ✉e-mail: d.altmann@imperial.ac.uk; r.boyton@imperial.ac.uk

a period of largely breakthrough infections by Omicron subvariants in vaccinated populations[5]. The UK, which collects relatively granular population data through the Office for National Statistics (ONS), estimates over 2-million with Long Covid in the UK alone, with US Census Bureau data estimating over 16-million cases there[1,6]. There are numerous hypotheses as to the immunopathogenesis of Long Covid[7]. Within a medical research agenda which has been strongly driven by the initiatives of the patients themselves[8], one area of focus has been the hypothesis that disease may have been triggered by anomalies in the immune response to SARS-CoV-2 during acute infection. It has been variously proposed either that Long Covid sufferers are unusual in having especially low adaptive immunity to the virus, or alternatively, that the persistent symptoms may be related to an excessively high and uncontrolled anti-viral response. The 'high anti-viral response' hypothesis is potentially compatible with a related hypothesis of Long Covid aetiology, namely that there is a chronic reservoir of persistent SARS-CoV-2 antigen, for example in the gut[9,10]. Several studies have looked at T cell subset phenotypes and at T cell immunity to SARS-CoV-2 comparing individuals with or without Long Covid finding a number of potential differences though, as yet, no consensus[11-18]. Some find evidence of enhanced SARS-CoV-2 adaptive immunity in those progressing to Long Covid: for example, among ongoing pulmonary Long Covid cases, substantially increased CD4 and CD8 responses were found[12], while another study showed a more sustained T cell and Ab response, albeit in a more severe cohort, many of whom had been hospitalised[13]. Increased convalescent antibody titers have been reported by some as a marker of Long Covid. Other cohort studies either found no difference between groups in SARS-CoV-2 immunity[15], or that reduced or rapidly declining responses were found in Long Covid[16-18].

Throughout the pandemic we have reported longitudinal immune parameters in the BARTS COVIDsortium London Healthcare worker (HCW) cohort, analysed since March 2020[19-24], and including proteomic analysis of Long Covid biomarkers[23]. That is, 731 HCW were recruited into the bioresource, including a cross-sectional case controlled sub-study of 136 HCW, 76 of whom had mild/asymptomatic SARS-CoV-2 infection during the first wave, captured by serial sampling, with SARS-CoV-2 infection determined by baseline and weekly nasal RNA swabs, Roche Cobas® SARS-CoV-2 reverse transcriptase polymerase chain reaction (RT-PCR) test and baseline and weekly S1 and N antibody (Ab) testing (see Methods). All HCW, irrespective of infection status, also reported data on a symptom questionnaire. While some Long Covid studies have been criticised in that they recruit 'self-reported' cases and also sometimes lack control populations, our study offers a number of advantages: HCW gave longitudinal blood samples allowing us to compare immune parameters in HCW with mild or asymptomatic laboratory confirmed SARS-CoV-2 infection during the first wave. Furthermore, the symptom diary questionnaires were initiated at a time when there was no knowledge of Long Covid and it's symptom profile, making this a study relevant to symptom persistence in Long Covid, collected in real-time at a period of the pandemic when HCW had no knowledge of the condition.

Here, we show that parameters of the longitudinal Ab and T cell response to SARS-CoV-2 antigens following mild or asymptomatic infection during the first wave are indistinguishable between HCW who did or did not go on to develop persistent symptoms.

## Results and discussion

Reviewing persistent symptoms at 6-months after infection in March 2020 during the first wave, there was a significant increase in reporting persistent symptoms in those HCW who had become infected with SARS-CoV-2 compared to those who had not. The former were more likely to report shortness of breath (8/91, (8.8%) vs 4/308, (1.3%);

$p = 0.0084$) (Table 1). By 12-months the questionnaire had been expanded to include additional symptoms. Again, the previously infected group were more likely to report a range of persistent symptoms including fatigue (18/86, (20.9%) vs 9/271, (3.3%); $p = 0.0023$), shortness of breath (10/86, (11.6%) vs 5/271, (1.8%); $p = 0.0023$), anxiety (11/86, (12.8%) vs 7/271, (2.6%); $p = 0.0046$) and insomnia (9/86, (10.5%) vs 5/271, (1.8%); p = 0.0069), the most frequent being fatigue (Table 2).

We were thus able to compare HCW who had suffered contemporaneous, first-wave, mild/asymptomatic COVID-19, with or without evidence of persistent symptoms, looking at pre-vaccination immunity with respect to: Ab binding response to spike (S) and nucleocapsid (N) during and following the acute infection, as well as neutralizing Ab titers and T cell responses at 4 months after infection (Fig. 1). This not only allowed us to investigate whether symptom persistence was associated with especially high or low parameters for these elements of SARS-CoV-2 adaptive immunity acutely or in subsequent months, it also allowed us to use the trajectory of the longitudinal N Ab response as a proxy measurement of whether there was likely to be an ongoing, persistent reservoir of antigen[9-11]. Antigen persistence might be predicted to correlate with a sustained or rising N Ab response; from first principles, a persistent antigen reservoir may be visible through its immunogenicity and impact on ongoing stimulation of the Ab response to N, and thus a tendency to increased levels. We initially considered whether the groups who did or did not go on to experience persistent symptoms assessed at 6-months differed in any aspects of their immune response to the virus (Fig. 1A–I). With respect to binding Ab to RBD, there was no difference between groups in terms of peak Ab titer, with Ab level similarly maintained in both groups through to week 24 (Fig. 1A). Each group contained a small minority of HCW with a low or undetectable Ab response, as we have previously described[24]. A similar pattern was also evident for the anti N Ab response (Fig. 1B). We did not find evidence for aberrant anti-viral immunity as a predictor of persistent symptoms and the data also were not strongly supportive either of differential immune waning or of ongoing immune stimulation from a persistent immune reservoir of virus.

Furthermore, we observed no significant differences between these groups when assayed at 16–18 weeks after acute infection for Ab neutralization IC50 against ancestral spike pseudovirus (Fig. 1D). ELISpot was used to look for any differences between T cell response frequency specific for any of: spike protein, N protein (Fig. 1E), spike mapped epitope pool (MEP)[19-21,24,25], N MEP, M MEP, or ORF3a/7a MEP (Fig. 1F). Mean responder cell frequency to each antigen was similar between groups, as was the absolute frequency of responding individuals. Each group contained a small minority of individuals mounting no detectable T cell response following infection, as we have previously described[24]. It has been proposed that EBV reactivation may be an underlying factor in Long Covid, so we here included T cell data from the response in each group to the CEF peptide pool, allowing us to use responses within the pool to common HLAI epitopes within the EBV immediate early gene products BMLF, BRLF and BZLF as well EBNA3 as a proxy for T cell recognition of reactivated EBV. No difference was seen between the persistent and recovery groups, although with the caveat that these responses also encompass CMV and influenza epitopes (Fig. 1G). We then explored longitudinal T cell responses in HCW with PCR-confirmed infection, who subsequently did or did not report persistent symptoms, and for whom weekly PBMC samples were available. In order to be sure that we had not missed T cell response differences related to selected epitopes within the viral proteome, these studies utilised full SARS-CoV-2 epitope megapools encompassing either spike or non-spike peptides (Fig. 1H, I; Supplementary Fig. 1)[26]. Again, no differential pattern of response between HCW with or without persistent symptoms could be seen. We thus found no evidence for either a differential neutralizing Ab response or

**Table 1 | Demographic and symptom data for HCW cohort at 6 months after SARS-CoV-2 infection**

|  | SARS-CoV-2 infection naive | SARS-CoV-2 infection (First UK wave) Recovered | SARS-CoV-2 infection (First UK wave) Persistent symptoms | p value Fishers exact (Bonferroni corrected) Symptom frequency infection naive vs infected |
|---|---|---|---|---|
| HCW: n (%) | 308 (77) | 75 (19) | 16 (4) |  |
| Mean age: years (range) | 39 (18–71) | 41 (21–62) | 38 (26–62) |  |
| Female: n (%) | 211 (69) | 47 (63) | 11 (69) |  |
| Male: n (%) | 97 (31) | 28 (37) | 5 (31) |  |
| Ethnicity: |  |  |  |  |
| White: n (%) | 213 (69) | 56 (75) | 10 (63) |  |
| Minority ethnic group (UK): n (%) | 95 (31) | 19 (25) | 6 (37) |  |
| Persistent symptom(s): n (%) |  |  |  |  |
| Fatigue | 7 (2) | 0 (0) | 8 (50) | 0.0602 |
| Cough | 6 (2) | 0 (0) | 0 (0) | 1.0 |
| Shortness of breath | 4 (1) | 0 (0) | 8 (50) | **0.0084** |
| Chest pain | 2 (1) | 0 (0) | 1 (6) | 1.0 |
| Myalgia | 4 (1) | 0 (0) | 0 (0) | 1.0 |
| Headache | 5 (2) | 0 (0) | 3 (19) | 1.0 |

*HCW* Healthcare worker. Bold formatting indicates a Bonferroni corrected statistically significant result.

a differential T cell response to any of the tested regions of the SARS-CoV-2 viral proteome between the group who would make a full recovery and the one that would experience persistent symptoms.

We reappraised the HCW cohort at 12 months after first wave acute infections, looking now at a more detailed symptom questionnaire and at immune parameters at this later timepoint, after two Pfizer vaccine doses. Fatigue was still the most common persistent symptom (Table 2). We identified no difference in the hybrid immunity Ab response to S1 RBD, or N in those with persistent symptoms compared to those that had fully recovered (Fig. 2A, B). This was also the case for neutralizing Ab IC50 and T cell responses to S and to N MEP. (Fig. 2C, D; Supplementary Fig. 2).

A limitation of this study is that the classification of first-wave COVID-19 infections that were either 'recovered' or 'persistent symptoms' used symptom diaries that pre-dated a fuller understanding of Long Covid and its defining symptoms[4]. In some respects it is also a strength of our study to have worked with a HCW cohort reporting their infection at a time when Long Covid was not yet recognised. Relative to subsequent bespoke cohorts designed to study Long Covid, our dataset is limited by the low number of individuals in our HCW cohort with persistent symptoms. Nevertheless, the conclusion from this comparison is that development of persistent symptoms is not explicitly correlated with a differential T cell or Ab response to acute viral infection. That is, there is no indication of an overt anomaly in handling of the initial viral infection. Furthermore, it is a *sine qua non* for other persistent or latent infections such as EBV that the persistent phase is evidenced by a lingering/rising Ab titer to a subset of antigens[27]. We note that some published data on Long Covid cohorts differed from our findings in identifying changes in SARS-CoV-2 adaptive immunity, but they are distinct from our study to the extent that these are studies either focused on more severe acute infections, or were analyzed at much later timepoints[11–18]. That we did not see a differential Ab response in the persistent symptoms group of this cohort might be considered to argue against the 'persistent antigen reservoir' hypothesis as commonly causal in Long Covid.

## Methods
This research complies with all relevant ethical regulations. The COVIDsortium Healthcare Workers bioresource was approved by the ethical committee of UK National Research Ethics Service (20/SC/0149) and registered on ClinicalTrials.gov (NCT04318314). The study conforms to the principles of the Helsinki Declaration with all subjects giving written and informed consent. The parent study outline and baseline characteristics of the prospective longitudinal HCW cohort established to study immune protection and pathogenesis in COVID-19 is available at Wellcome Open Res 2020, 5:179 https://doi.org/10.12688/wellcomeopenres.16051.1.

### Study cohort
The COVIDsortium Healthcare Worker (HCW) cohort and details of subsequent follow-up and analysis have been described in detail elsewhere[24]. In brief, adult HCW (defined as >18 y) were invited to participate via local advertisement (see https://covid-consortium.com). A cohort (n = 400) was initially recruited from St Bartholomew's Hospital, London in the week of 23rd–31st March 2020. Recruitment was then extended (27th April-7th May 2020) to include 331 additional participants from: St Bartholomew's Hospital (n = 101), NHS Nightingale Hospital (n = 10), and Royal Free Hospital, London (n = 220) making 731 HCW in total (Supplementary Fig. 3).

A prospective, observational, longitudinal cohort design was used that included questionnaires collecting data about demographic information, symptoms and exposure risk. Samples were collected at baseline and at weekly follow-up visits for 15 weeks, and at 4, 6, and 12 months. At weekly follow-up visits, symptom burden information was recorded using a standardized questionnaire recording symptoms classified as 'case-definition' (fever, new continuous dry cough or a new loss of taste or smell), 'non-case-definition' (symptoms other than case-defining symptoms), or no symptoms reported. When HCW returned from symptomatic self-isolation, convalescent samples were collected. Laboratory confirmed SARS-CoV-2 infection during the first wave was identified by baseline and weekly, RT-PCR using Roche Cobas® SARS-CoV-2 test and IgG Ab assay to spike protein S1 antigen, (Euroimmun Anti-SARS-CoV-2 enzyme-linked immunosorbent assay [ELISA] #EI2606-9601G); and anti-nucleocapsid total Ab assay (Roche Elecsys Anti-SARS-CoV-2 electrochemiluminescence-immunoassay [ECLIA] #09203079190). Ab ratios > 1.1 were considered positive for the Euroimmun SARS-CoV-2 ELISA and >1 was considered positive for the Roche Elecsys anti-SARS-CoV-2 ECLIA. A total of 157 (21.5%) HCW had laboratory confirmed SARS-CoV-2 infection of whom 49 (31%)

**Table 2 | Demographic and symptom data for HCW cohort at 12 months after SARS-CoV-2 infection**

| | SARS-CoV-2 infection naive | SARS-CoV-2 infection (First UK wave) Recovered | SARS-CoV-2 infection (First UK wave) Persistent symptoms | p value Chi squared (Bonferroni corrected) Symptom frequency infection naive vs infected |
|---|---|---|---|---|
| HCW: n (%) | 271 (76) | 61 (17) | 25 (7) | |
| Mean age: years (range) | 40 (18–69) | 42 (21–62) | 39 (25–59) | |
| Female: n (%) | 178 (66) | 40 (66) | 18 (72) | |
| Male: n (%) | 93 (34) | 21 (34) | 7 (28) | |
| Ethnicity: | | | | |
| White: n (%) | 192 (71) | 48 (79) | 17 (68) | |
| Minority ethnic group (UK): n (%) | 79 (29) | 13 (21) | 8 (32) | |
| Persistent symptom(s): n (%) | | | | |
| Anxiety | 7 (2.7) | 0 (0) | 11 (44) | **0.0046** |
| Sore throat | 0 (0) | 0 (0) | 3 (12) | **0.0046** |
| Low mood | 9 (3.4) | 0 (0) | 10 (40) | 0.0644 |
| Insomnia | 5 (1.9) | 0 (0) | 9 (36) | **0.0069** |
| Headache | 10 (3.8) | 0 (0) | 10 (40) | 0.1219 |
| Pins and Needles | 3 (1.1) | 0 (0) | 4 (16) | 0.8947 |
| Confusion | 4 (1.5) | 0 (0) | 2 (8) | 1.0 |
| Fainting | 2 (0.7) | 0 (0) | 0 (0) | 1.0 |
| Dizziness | 4 (1.5) | 0 (0) | 4 (16) | 1.0 |
| Weight loss | 7 (2.7) | 0 (0) | 0 (0) | 1.0 |
| Loss of appetite | 3 (1.1) | 0 (0) | 1 (4) | 1.0 |
| Abdominal pain | 3 (1.1) | 0 (0) | 1 (4) | 1.0 |
| Nausea or vomiting | 2 (0.7) | 0 (0) | 0 (0) | 1.0 |
| Constipation | 1 (0.4) | 0 (0) | 3 (12) | 0.3818 |
| Diarrhoea | 0 (0) | 0 (0) | 1 (4) | 1.0 |
| Fatigue | 9 (3.4) | 0 (0) | 18 (72) | **0.0023** |
| Joint ache or swelling | 6 (2.3) | 0 (0) | 8 (32) | 0.0736 |
| Muscle aches | 8 (3.0) | 0 (0) | 8 (32) | 0.3013 |
| Palpitations | 4 (1.5) | 0 (0) | 4 (16) | 1.0 |
| Pain on breathing | 0 (0) | 0 (0) | 1 (4) | 1.0 |
| Shortness of breath | 5 (1.9) | 0 (0) | 10 (40) | **0.0023** |
| Chest tightness | 3 (1.1) | 0 (0) | 1 (4) | 1.0 |
| Cough | 2 (0.7) | 0 (0) | 2 (8) | 1.0 |

HCW Healthcare worker. Bold formatting indicates a Bonferroni corrected statistically significant result.

were asymptomatic. Infections were asymptomatic or mild. The HCW cohort is, therefore, a working age, longitudinal community rather than hospitalized COVID-19 cohort with approximately synchronous infection during the first wave that peaked on or around March 23rd, 2020.

At 12-months follow up, HCW were asked to complete a symptom questionnaire that enabled subsequent assessment of symptom persistence. Anti-nucleocapsid antibody titers were used to define HCW participants as infection naive (n = 271) or having a history of previous SARS-CoV-2 infection (n = 86). Questionnaire data was used to establish if HCW that had a history of SARS-CoV-2 infection during the first wave had either Recovered (n = 61) or had Persistent symptoms (n = 25). The demographic and symptom characteristics of each of these three groups are shown in Table 2.

## Quantification of SARS-CoV-2 specific binding antibodies

SARS-CoV-2 Ab testing was carried out at UK Health Security Agency, Porton Down: Anti-nucleocapsid and anti-spike antibody detection testing was conducted using the Roche cobas® e801 analyser. Anti-nucleocapsid antibodies were detected using the qualitative Roche

Elecsys® anti-SARS-CoV-2 electrochemiluminescence immune analyzer (ECLIA) nucleocapsid assay (Roche ACOV2, #09203079190). Anti-RBD antibodies were detected using the quantitative Roche Elecsys® anti-SARS-CoV-2 ECLIA spike assay (Roche ACOV2S, #09289275190). Assays were performed and calibrated as recommended by the manufacturer. Anti-Nucleocapsid results are expressed as a cutoff index (COI) value based on the electrochemiluminescence signal of a two-point calibration, with results COI ≥ 1.0 classified as positive. Anti-S1 RBD results are expressed as units per ml (U/ml) similarly based on a two-point calibration and a reagent specific master curve, with a quantitative range of 0.4 to 2500 U/ml. Samples with a value of ≥1.0 U/ml are interpreted as positive for spike antibodies.

## SARS-CoV-2 ancestral strain pseudotype virus microneutralization assays

SARS-CoV-2 pseudotype neutralization assays were conducted using pseudotyped lentiviral particles prepared by linear polyethylenimine 25 K (Polysciences #23966) co-transfection of HEK-293T (ATCC, #CRL-3216) with SARS-CoV-2 spike pcDNA expression plasmid, HIV gag-pol

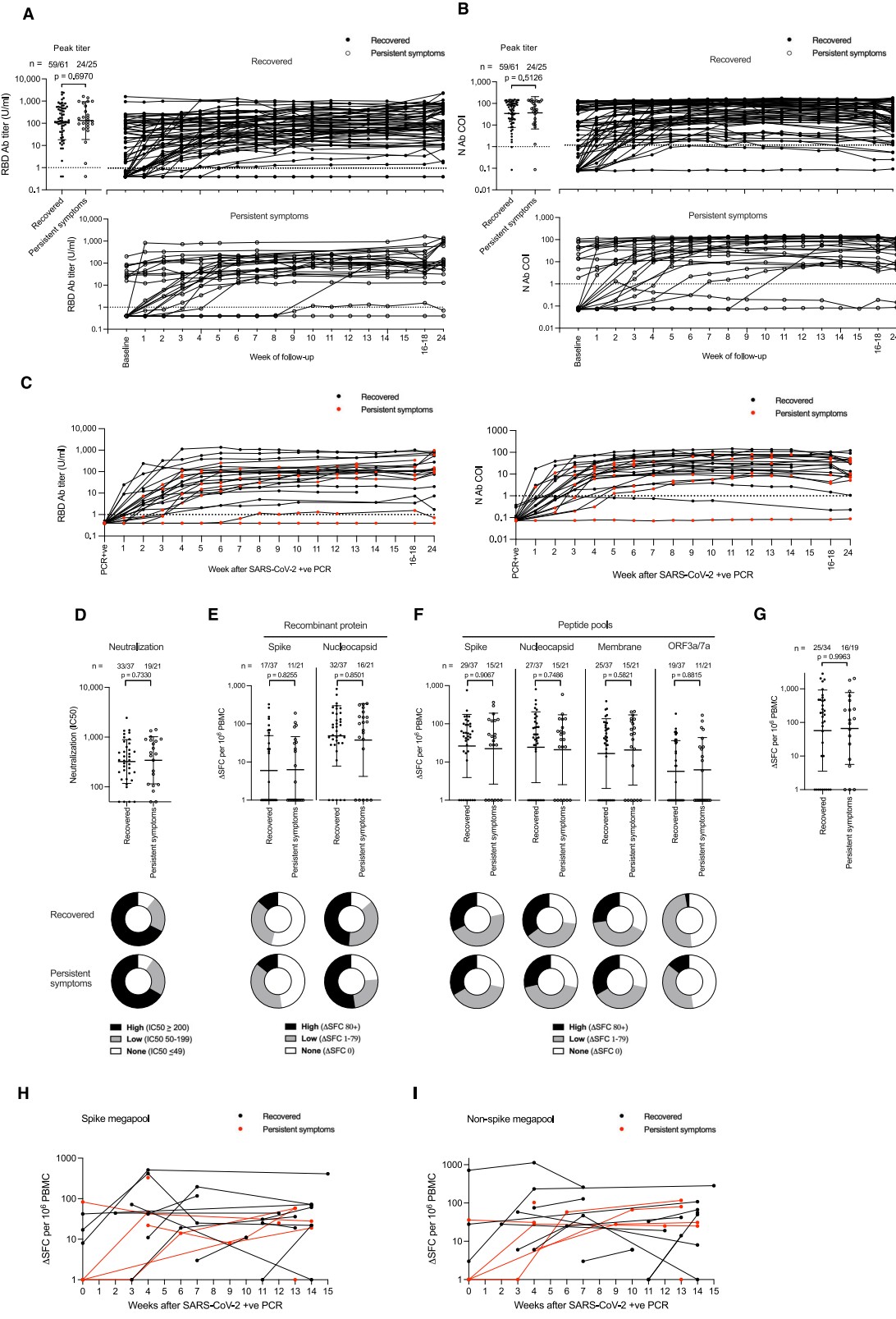

p8.91 plasmid and firefly luciferase expressing plasmid pCSFLW at a 1:1:1.5 ratio. TCID assays were performed by transduction of Huh7 cells (ECACC, #01042712) to calculate the viral titer and infectious dose for subsequent neutralization assays.

To perform pseudotype virus neutralization assays, study participant serum was heat-inactivated at 56 °C for 30 min to remove complement activity. Serum was diluted in DMEM and a 100 μl 7 point, 2-fold dilution series performed in duplicate in white, flat-bottom 96-well plates (ThermoFisher, #136101) with a starting dilution of 1:20. In total $1 \times 10^5$ Relative light units (RLU) of SARS-CoV-2 pseudotyped lentiviral particles were added to each well and incubated at 37 °C for 1 h. Eight control wells per plate received pseudotype and cells only (virus control) and 8 wells received cells only (background control). Negative controls of pooled pre-pandemic

**Fig. 1 | Longitudinal antibody and T cell responses to SARS-CoV-2 during the first 16–18 weeks after infection in HCW that either recovered or reported persistent symptoms.** A cohort of health care workers (HCW) recruited in March 2020 with laboratory confirmed SARS-CoV-2 infection ($n = 86$, 33% male) were followed up weekly. Serum was assayed for **A** S1 RBD and **B** Nucleocapsid antibody titers. Twenty-five HCW reported persistent symptoms at 12 months post-infection (open circle, 28% male) whilst the remaining 61 HCW fully recovered (closed circle, 34% male). In both panels, peak antibody titer during the 24-week follow-up and longitudinal data is plotted for each HCW. **C** S1 RBD and nucleocapsid antibody titers plotted longitudinally relative to timing of positive SARS-CoV-2 PCR (Recovered, black, $n = 15$, 40% male; Persistent symptoms, red, $n = 9$, 44% male). Thirty-seven of Recovered HCW (closed circle, 30% male) and 21 of the HCW with persistent symptoms (open circle, 24% male) were assayed for **D** Neutralizing antibody (IC50) against SARS-CoV-2 ancestral (Wuhan Hu-1) strain pseudovirus at 16–18 weeks; **E** T cell response against spike and nucleocapsid recombinant proteins and **F** peptide pools containing mapped epitopes from spike, nucleocapsid,

membrane and ORF3a/7a proteins. **G** T cell response against peptide pool containing epitopes from Cytomegalovirus, Epstein Barr virus and Influenza (Recovered, $n = 34$, closed circle, 29% male; Persistent symptoms, $n = 19$, open circle, 26% male). For **A**–**G**, numbers of HCW showing a positive response for each assay are shown at the top of each plot and the proportion of each group with a high (black), low (grey) or no response (white) are represented by doughnut plots below. Longitudinal T cell responses to spike **H** and non-spike **I** peptide megapools assayed by ELISPOT (in Recovered HCW, black circle, $n = 14$, 36% male; and HCW reporting Persistent symptoms, red circle, $n = 8$, 50% male) at 12 months after SARS-CoV-2 infection. Data is plotted relative to when HCW had a SARS-CoV-2 positive PCR. Two-tailed Mann-Whitney U tests (Graphpad Prism version 8.0) were used to test for significant differences between recovered HCW and those reporting persistent symptoms. Source data are provided as a Source Data file. Error bars shown are geometric mean ± 1 geometric SD. Ab antibody, COI cut-off index, N nucleocapsid, ORF open reading frame, PBMC peripheral blood mononuclear cells, RBD receptor binding domain, SD standard deviation, SFC spot forming cells, U units.

sera, collected prior to 2008, and a positive neutraliser were spaced throughout the plates. RLUs for each well were standardised against technical positive (virus control) and negative (cells only) controls on each plate to determine percentage neutralization. In total $4 \times 10^4$ Huh7 cells suspended in 100 µl media were added/well and incubated for 72 h at 37 °C and 5% $CO_2$. Firefly luciferase luminescence was measured using Steady-Glo® Luciferase Assay System (Promega #E2510) and a CLARIOStar Plate Reader (BMG Labtech). Average neutralization of duplicates was calculated for each serum dilution. Neutralization curves for each serum sample were plotted and the percentage neutralization modelled as a logistic function of the serum dilution factor (log10).

SARS-CoV-2 ancestral strain live virus microneutralization assays The SARS-CoV-2 strain 2019-nCoV/BavPat1/2020 (Wuhan Hu-1) virus isolate was obtained from the European Virus Archive Global (EVAg, Charité Universitätsmedizin Berlin, Germany). Virus stocks were prepared by inoculation of VeroE6 seeded 75cm² cell culture flasks with virus cell culture supernatant containing $2.2 \times 10^6$ Plaque forming units (PFU) in a volume of 10 ml DMEM containing 10% FBS. Virus-containing culture medium was harvested when >80% of cells showed cytopathic effect (CPE).

Titers of the viral stocks were determined by challenging monolayers of VeroE6 (ATCC, #VERO C1008) cells with serial dilutions of virus and culturing for 20 h before in situ intracellular staining to identify foci of infection. Staining was performed by fixing cells with ice-cold methanol:acetone (50:50) and incubating with convalescent sera (1:2000 dilution) in PBS with 1% FBS for 1 h at 37 °C. Following 3 PBS washes cells were incubated with a goat anti-human IgG β-galactosidase-conjugated antibody (1:400 dilution, Polyclonal, Southern Biotech #2040-06) for a further 1 h at 37 °C. Cells were washed 3 more times with PBS before incubation with 300 µl of 0.5 mg/ml 5-bromo-4-chloro-3-indolyl β-D-galactopyranoside chromogenic substrate (X-gal, Bioline, #BIO-37035) in PBS containing 3 mM potassium ferricyanide and 1 mM magnesium chloride at 37 °C for up to 4 h. Infected cells stained blue and were counted as foci of infection to determine the virus titer expressed as focus forming units (FFU) per ml.

To perform live virus neutralization assays, VeroE6 cells were seeded in 96-well plates 24 h before infection. Titrations (7 point 2-fold dilution series) of heat-inactivated participant sera were set up in duplicate, starting at a 1:20 dilution and incubated with $3 \times 10^4$ FFU of SARS-CoV-2 virus (TCID100) at 37 °C for 1 h. Serum/virus preparations were added to cells and incubated for 72 h. Surviving cells were fixed in formaldehyde and stained with 0.1% (w/v) crystal violet solution was resolubilized in 1% (w/v) sodium dodecyl sulphate solution. Absorbance readings were taken at 570 nm using a CLARIOStar PlateReader (BMG Labtech). Negative controls of pooled pre-pandemic sera (collected before 2008) and pooled serum from neutralization-positive

SARS-CoV-2 convalescent individuals were spaced across the plates. Absorbance for each well was standardized against technical positive (virus control) and negative (cells only) controls on each plate to determine percentage neutralization values. IC50 values were determined from neutralization curves. All authentic SARS-CoV-2 propagation and microneutralization assays were performed in a containment level 3 facility.

## T cell assays

Peripheral blood mononuclear cells (PBMC) were isolated from heparinized blood samples using Pancoll (Pan Biotech #P04-60500) or Histopaque®-1077 Hybri-Max™ (Sigma-Aldrich #H8889) density gradient centrifugation in SepMate™ tubes (Stemcell #85450). Thirteen-20mer peptides based on the protein sequences of SARS-CoV-2 S1 (spike), nucleocapsid (N), membrane (M) or open reading frames 3a and 7a (ORF3a/7a) described previously were synthesized (GL Biochem Shanghai Ltd, China)[24,25]. SARS-CoV-2 S1 spike and nucleocapsid recombinant proteins were obtained from the Centre for AIDS Reagents (CFAR), National Institute for Biological Standards and Control (NIBSC), UK and were from Dr Peter Cherepanov, Francis Crick Institute, UK. To stimulate PBMC, recombinant proteins, separate pools of sequences for Spike (18 peptides), N (10 peptides), M (6 peptides) and ORF3a/7a (7 peptides)[24,25], or a megapool of peptides covering the whole sequence of spike (127 peptides each of 20aa long and overlapping by 10aa) and a megapool of non-spike peptides (284 peptides)[26] were used. Assay PBMC were cultured in RPMI medium (Gibco #11875093) supplemented with 10% FBS, 1 x penicillin and streptomycin solution (Gibco #15140122) and 2 mM L-Glutamine (Gibco #A2916801) (R10). Pre- coated ELISpot plates (Mabtech #3420-2APT) were washed x4 with sterile PBS and were blocked with R10 for 1 h at room temperature. 200,000 PBMC were used per well and stimulated for 20 h with SARS-CoV-2 recombinant proteins (10 µg/ml), mapped epitope pools (10 µg/ml/peptide) or megapools (2ug/ml/peptide). Internal plate controls were R10 alone (without cells) and anti-CD3 (1:1000 dilution, clone CD3-2, Mabtech, #3605-1-50). Plates were developed with human biotinylated IFNγ detection Ab, directly conjugated to alkaline phosphatase (clone 7-B6-1-ALP, Mabtech, #3420-9 A), diluted 1:200 in PBS with 0.5% FBS, incubating 100 µl/well for 2 h, followed by 100 µl/well of sterile filtered BCIP/NBT-plus Phosphatase Substrate (Mabtech #3420-2APT) for 5 min. Plates were washed, dried, and read on an AID-ELISpot plate reader. Analysis of ELISpot data was performed in Microsoft Excel. The average of two R10 wells was subtracted from all peptide stimulated wells and any response that was less than 2 SD of the sample specific control wells was not considered peptide-specific. Results were expressed as difference in (delta) spot forming cells/10⁶ PBMC between the negative control and peptide stimulation. Results were excluded if negative control wells had >100 SFU/10⁶ PBMC or if

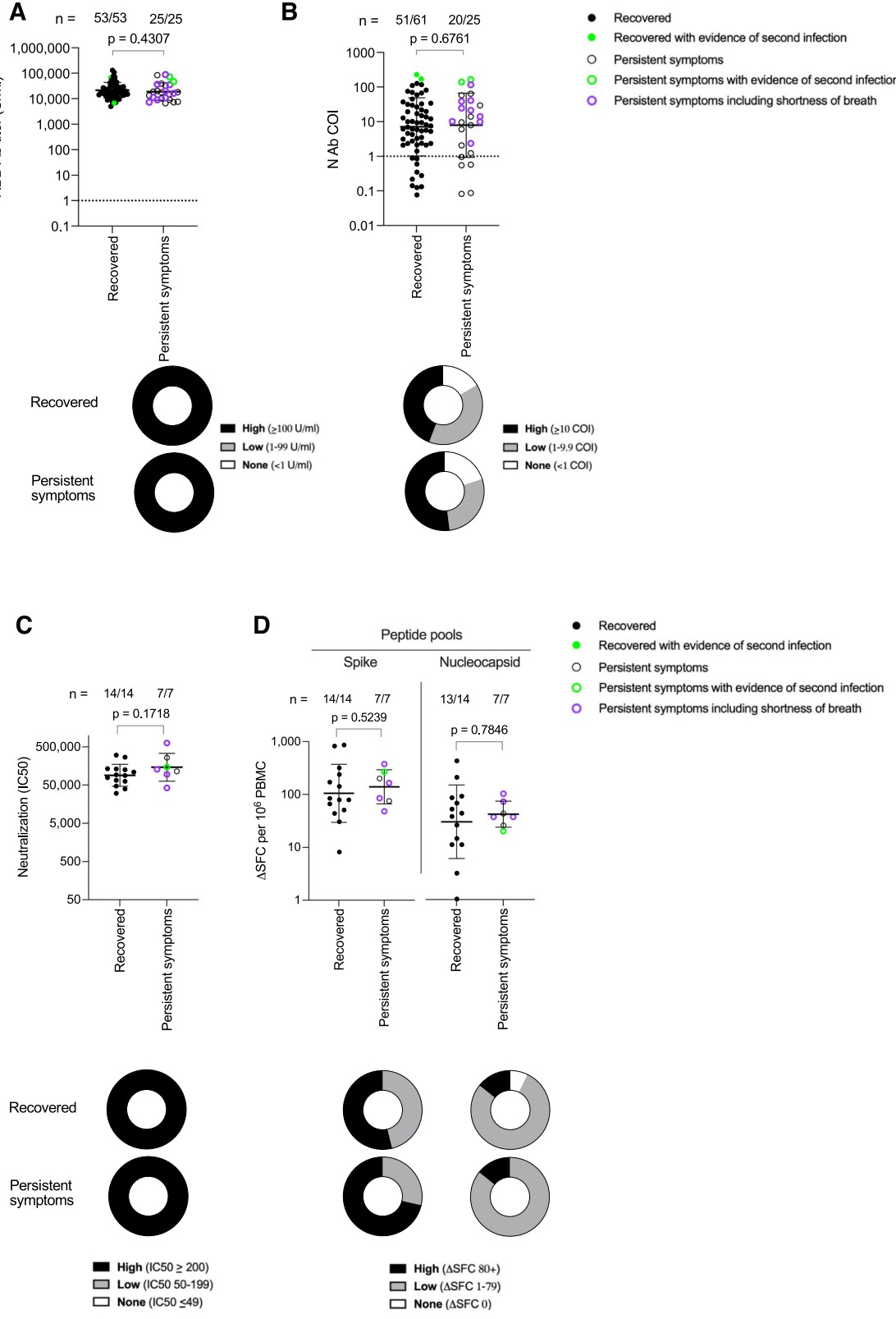

positive control wells were negative. Results were plotted using Prism v8.0 for Mac OS (GraphPad).

**Statistics and reproducibility**

Statistically significant differences in symptom frequency between infection naive and SARS-CoV-2 infected HCW were tested by Bonferroni corrected Fishers exact (Table 1) or Chi squared (Table 2) tests. Statistically significant differences between recovered and persistent symptom groups for T cell and antibody data were tested by two-tailed Mann-Whitney U test (Figs. 1, 2, Supplementary Figure 2) as data was assumed to have a non-Gaussian distribution and therefore two-tailed nonparametric tests were used. A p value of <0.05 was considered significant. Prism v. 8.0 for Mac was used for analysis.

**Fig. 2 | Antibody and T cell responses against SARS-CoV-2 at 12 months after laboratory confirmed infection in HCW that either Recovered or reported Persistent symptoms. A** S1 RBD (Recovered, closed circle, $n = 53$, 40% male); Persistent symptoms, open circle, $n = 25$, 28% male) and **B** Nucleocapsid antibody (Recovered, closed circle, $n = 61$, 34% male; Persistent symptoms, open circle, $n = 25$, 28% male) titers in HCW 12 months after laboratory confirmed SARS-CoV-2 infection. S1 RBD data is plotted for HCW who had received 2 doses of COVID-19 vaccine at the time of the 12-month blood sample. Numbers of HCW with a positive antibody titer are shown at the top of each plot. The proportion of HCW with a high (black), low (grey) or no antibody (white) titer is shown in doughnut plots below. Fifteen Recovered HCW (closed circle, 57% male) and 7 HCW with Persistent symptoms (open circle, 57% male) were assayed for **C** Neutralizing antibody IC50 against live ancestral (Wuhan Hu-1) SARS-CoV-2 virus and **D** T cell responses against spike and nucleocapsid mapped epitope peptide pools, 12 months after laboratory

confirmed SARS-CoV-2 infection. Numbers of HCW showing a positive neutralizing antibody or T cell response for each assay are shown at the top of each plot. The proportion of HCW in each group with a high (black), low (grey) or no response (white) (as defined in each key) are represented by doughnut plots below. Two-tailed Mann-Whitney U test (Graphpad Prism version 8.0) was used to test for significant differences between Recovered HCW and those reporting Persistent symptoms. Source data are provided as a Source Data file. HCW with serological evidence of a subsequent SARS-CoV-2 re-infection between 6 and 12 months are annotated in green. HCW who reported persistent shortness of breath at 12 months are shown as purple open circles. Error bars shown are geometric mean $\pm$ 1 geometric SD. Ab antibody, COI cut-off index, HCW health care workers, N nucleocapsid, PBMC peripheral blood mononuclear cells, RBD receptor binding domain, SD standard deviation, SFC spot forming cells, U units.

## Reporting summary

Further information on research design is available in the Nature Portfolio Reporting Summary linked to this article.

## Data availability

All data needed to evaluate the conclusions of this paper are presented in the paper or the Supplementary Material. Source data are provided with this paper in the Source Data file. Source data are provided with this paper.

## Code availability

No custom computer code or algorithm was used to generate the results that are reported in the paper.

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

## Acknowledgements

The authors thank HCW participants for participating in the study and the research teams involved in recruitment, obtaining consent, and sampling the HCW participants. The COVIDsortium Healthcare Workers bioresource is approved by the ethical committee of UK National Research Ethics Service (20/SC/0149) and registered on Clinical-Trials.gov (NCT04318314). The study conforms to the principles of the Helsinki Declaration, and all subjects provided written informed consent. The SARS-CoV-2 ancestral strain pseudotyped lentivirus was

supplied by Nigel Temperton. This work is licensed under a Creative Commons Attribution 4.0 International (CC BY 4.0) license, which permits unrestricted use, distribution, and reproduction in any medium, provided the original work is properly cited. To view a copy of this license, visit https://creativecommons.org/licenses/by/4.0/. This license does not apply to figures/photos/artwork or other content included in the article that is credited to a third party; obtain authorization from the rights holder before using such material. RJB and DMA are supported by MRC (MR/S019553/1, MR/R02622X/1, MR/V036939/1, MR/W020610/1), NIHR Imperial Biomedical Research Centre (BRC):IT-MAT, Cystic Fibrosis Trust SRC (2019SRC015), NIHR EME Fast Track (NIHR134607), NIHR Long Covid (COV-LT2-0027), Innovate UK (SBRI 10008614) and Horizon 2020 Marie Skłodowska-Curie Innovative Training Network (ITN) European Training Network (No 860325). ÁM is supported by MRC (MR/W020610/1), NIHR EME Fast Track (NIHR134607), Rosetrees Trust, The John Black Charitable Foundation, and Medical College of St Bartholomew's Hospital Trust. The COVID-sortium is supported by funding donated by individuals, charitable Trusts, and corporations including Goldman Sachs, Kenneth C Griffin, The Guy Foundation, GW Pharmaceuticals, Kusuma Trust, and Jagclif Charitable Trust, and enabled by Barts Charity with support from UCLH Charity. Wider support is acknowledged on the COVIDsortium website. Institutional support from Barts Health NHS Trust and Royal Free NHS Foundation Trust facilitated study processes, in partnership with University College London and Queen Mary University of London. MKM is supported by UKRI/NIHR UK-CIC, Wellcome Trust Investigator Award (214191/Z/18/Z) and CRUK Immunology grant (26603). JCM, CM and TAT are directly and indirectly supported by the University College London Hospitals (UCLH) and Barts NIHR Biomedical Research Centres and through the British Heart Foundation (BHF) Accelerator Award (AA/18/6/34223). TT is funded by a BHF Intermediate Research Fellowship (FS/19/35/34374). MN is supported by the Wellcome Trust (207511/Z/17/Z) and by NIHR Biomedical Research Funding to UCL and UCLH. The funders had no role in study design, data collection, data analysis, data interpretation, or writing of the report.

## Author contributions

R.J.B & D.MA. conceptualised and designed the study reported. R.J.B and D.M.A designed and supervised the T cell experiments. Á.M. designed and supervised the nAb experiments. T.B. and A.Se supervised whole spike and S1 RBD IgG and N IgG/IgM studies. C.J.R. developed, performed and analyzed T cell experiments. L.S performed the T cell experiment using the CEF peptide pool supervised by M.K.M. J.M.G. and C.P developed, performed and analyzed nAb experiments. A.D.O performed and analyzed spike, RBD and N antibody assays. T.B., C.M., Á.M., T.A.T., J.C.M., and M.N established the COVIDsortium HCW cohort.

C.J.R., D.M.A., and R.J.B. analyzed the data. C.J.R., G.J., J.M.G., C.P., A.D.O., C.M., T.A.T., J.C.M., M.K.M., A.Se., T.B., M.N., Á.M., D.M.A., and R.J.B. interpreted the data. R.J.B. and D.M.A wrote the manuscript with input from the authors. All authors reviewed and edited the manuscript and figures.

## Competing interests

R.J.B. and D.M.A. are members of the Global T cell Expert Consortium and have consulted for Oxford Immunotec outside the submitted work. DMA has received honorarium payments from Pfizer, AstraZeneca and Novavax for consultancy work. The remaining authors declare no competing interests.

## Additional information

## COVIDsortium investigators

Daniel M. Altmann [1]✉, Catherine J. Reynolds[2], George Joy[3,4], Ashley D. Otter [5], Joseph M. Gibbons [6], Corinna Pade[6], Leo Swadling [7], Mala K. Maini [7], Tim Brooks[5], Amanda Semper[5], Áine McKnight[6], Mahdad Noursadeghi [7], Charlotte Manisty [3,4], Thomas A. Treibel[3,4], James C. Moon[3,4] & Rosemary J. Boyton [2,8]✉

A list of members and their affiliations appears in the Supplementary Information.

