## [Peer Review File · Nature Communications]

REVIEWER COMMENTS

Reviewer #1 (Remarks to the Author):

In this manuscript, Altmann et al leverage samples collected from an exceptionally well-characterized and well-controlled longitudinal cohort of healthcare workers enrolled during the first wave of SARS-CoV-2 transmission in March 2020. Immune responses to SARS-CoV-2 were assessed to determine if persistent symptoms after the resolution of acute SARS-CoV-2 infection (long COVID) were associated with dysregulated or sub-optimal immune responses. No statistically significant difference in the anti-Spike RBD antibody titers, N antibody titers, neutralizing antibody titers, or the frequency of IFN-g producing cells responding to Spike/N/M/Orf3/Orf7 peptide stimulation were observed. While these results are clear, there are several limitations of the study that dampen enthusiasm for the manuscript in its current form:

- 1) Many of the claims in the manuscript are quite broad and insufficient data are provided to support the expansive assertions. While there were no differences observed between the study groups analyzed in this manuscript, the immunologic assays are limited to antibody binding, neutralizing titers, and IFN-g ELISPOT analysis with 4 peptide pools spanning only a portion of the SARS-CoV-2 proteome. Fc-dependent/non-neutralizing antibody responses (such as ADCC/ADNP/ADCP activity) are not addressed in this analysis, and the IFN-g ELISPOT assay used in this analysis preferentially captures the cytotoxic CD8+ T cell response rather than the helper CD4+ T cell response. This is notable as SARS-CoV-2 infection and mRNA vaccination has been repeatedly shown to induce a more pronounced CD4 T cell response against Spike. These gaps do not invalidate any of the data presented, but it is overly broad to claim that there is no difference in “immunity” when only a portion of the potential immune response is analyzed. More precise language in the title and abstract could easily address these concerns.
- 2) Different numbers of individuals appear to have been analyzed in the different serological and cellular assays. It is unclear how these individuals were selected from the total study population and what the overlap is between the different assays.
- 3) The use of N Ab responses as a proxy measurement of an ongoing/persistent infection is unconventional. The paper cited to support this claim is a study on SARS-CoV-2 vaccine responses and does not mention antigen persistence. The studies that have posited that persistent SARS-CoV-2 antigen influences the stability and maturation of the long-term anti-SARS-CoV-2 immune response have primarily focused on the stability and accumulation of somatic hypermutations in memory B cells.

Reviewer #2 (Remarks to the Author):

This study attempts to determine if persistent symptoms after SARS-CoV-2 infection (Long-Covid) are associated with differential SARS-CoV-2 antibody or T cell responses by longitudinal analysis of healthcare workers with and without persistent symptoms. The study has some strengths. The Authors have access to a large cohort through the BARTS COVIDsortium with serial samples and weekly SARS-CoV-2 RT-PCR testing. Additionally, subjects provided a symptom diary initiated before anyone had knowledge of Long Covid, thus providing a potentially unbiased account of symptoms and retrospective grouping of subjects. However, enthusiasm is greatly tempered by the concerns outlined below.

Major Concerns:

-The antibody data presented here is consistent with other published studies, but the T cell data is not. There are concerns regarding the methodology used to measure T cell responses to SARS-CoV-2. The peptide pools used were only a fraction of the full proteins examined. For example, only 18 peptides (13-20mers) were used for the Spike protein. Most studies use 170+ overlapping 15mers peptides which encompass the whole Spike protein for T cell assays. It is unclear how these peptides were chosen and how they compare to stimulations with the full set of overlapping peptides. While it is noted on page 17 this was previously described no reference is provided. The same concern about the use of a limited number of peptide holds for Nucleocapsid, Membrane and ORF3a/7a proteins. Comparison of the limited pool of peptide used in this study to full set of overlapping peptides should be shown. The use of a limited set of peptides may result in incomplete evaluation of the actual T cell responses. The T cell responses also seem quite low compared to what is published by other groups. This is likely because the peptide pools are so small which brings up concerns about the accuracy of T cell assay. A longitudinal analysis of SARS-CoV-2-specific T cells, as was done for antibodies, would be useful and would shed light on the kinetics of the T cell response in the two groups.

-Key references which demonstrate T cell responses are maintained in individuals with Long Covid are not included and should be added and discussed (File JK et al. - JCI Insights and Littlefield K et al. - PLoS Pathogens). These studies show that SARS-CoV-2 specific T cells are maintained and significantly elevated compared to individuals who cleared the virus and had no persistent symptoms 4 weeks post infection, contradicting the data presented in this study.

-It would have been useful to follow the subjects out longer in Figure 1. The the antibody responses have not begun to wain in either cohort as shown in Figure 1 A+B. Figure 2 C+D does provide some useful T cell data on a very small subset of subjects (only 7 subject with persistent symptoms) at 12 months post infection and those with persistent symptoms have higher neutralization and higher number of SAR-CoV-2 specific T cells, although not statistically significant (please show the P value in the

figure). Since the 12 month time point maybe the most relevant a larger cohort of subjects with persistent symptoms is needed.

-Long haul symptoms are very heterogenous and can be broken down into 3 main categories - cardiovascular, pulmonary and neurological. This paper lumps them all together but other studies show that the various persistent symptoms may have different causes. The cohort is likely too small to subdivide by symptom type, but this could be a useful analysis, at least for data in Figure 1 where the number of subjects is fairly large. This should also be discussed.

Minor concerns:

-Has the group compared those with respiratory symptoms to those without in the persistent symptom group? Based on both tables about half of those with persistent symptoms had respiratory complications.

-Figure 1A and B show peak Ab titer. Was the titer at week 1 or 2 examined? Speed of response would be interesting with this dataset.

-Figure 2 – was there tracking to ensure there were no other illnesses or subsequent SARS-CoV-2 infection for those who “Recovered”

-It states on page 4 that an advantage of this study is the capture of PCR negative controls, but none of that data is shown. How do these data compare to PCR- responses?

-How were the cutoff for positivity determined for both antibody and T cell responses?

-How were the samples used in Figure 2 C and D chose from the larger cohort? Are they representative of the larger cohort? Data from Figure 1 should be used to show they are representative.

-There is discussion of EBV reactivation as an underlying factor for Long Covid. The authors say they assayed EBV specific T cell responses and found no difference. They can't say this because the CEF peptide pool is a mixture of CMV, EBV and Flu peptide and they are unable to determine which peptides are eliciting the response.

-Why wasn't a specific incubation time used for the T cell assay. It states they were incubated for 18-22 hours. It should have been the same duration for all assays.

-Showing the proportion of the response in Figure 1 and 2 that were high, low or none seems arbitrary. How were these cutoffs determined? These are not particularly helpful and could be removed.

-While subjects provide a self-assessment of symptoms was there any clinical diagnosis of Long Haul disease done at 12 months post infection?

-Does the group have any data on masking? Assumed high adherence in HCW population.

Reviewer #3 (Remarks to the Author):

The manuscript is a follow-up study of a very interesting cohort of health care workers in the UK who have been followed since before SARS-CoV-2 infection in March 2022, whereby COVID symptoms could be followed from before infection and longitudinally. The authors have used standard immunological assays to confirm infection, and at follow-up 4 months post infection, and later up to 12 months. In this paper an effort has been made to categorize the HCWs in long COVID participants, fully recovered, and uninfected participants. The main conclusion is that no immunological markers linked to long COVID symptoms were found, and they conclude that differences in adaptive immunity are unlikely contributors of long COVID.

Som questions need to be addressed:

1. The authors claim that the hypothesis of an immunopathogenesis is largely driven by patients, while published papers supporting such a hypothesis are not referenced at all, except a paper where the authors are themselves involved, identifying a proteome profile differing between long haulers and recovered, in the same patient cohort. The literature on differential immune responses in long COVID patients and recovered needs to be referenced. Long term immunological differences between long COVID and recovered can be found in Peluso MJ et al Cell Rep 2021 and in Phetsouphanh C et al Nat Immunol 2022.

2. Literature showing an association between low acute immune responses (Garcia-Abellan J et al J Clin Immunol 2021, Lerum TV et al Sci Rep 2021) or high peak convalescent antibodies (Blomberg B et al Nat Med 2021) are interesting in relation to the present study, and raises the question of whether later time point such as the 4 month time point, with potential waning immune responses, used in this study, can explain differences in findings. Note also Cervia C et al Nature Comm 2022. This needs to be discussed.

3. The main weakness of the study is the low sample size. With only 25 long COVID patients, where even one of the most persistent symptoms which is cognitive symptoms, such as memory and concentration problems, are not addressed, makes firm conclusions hard to make, and the conclusions should be modified accordingly.

4. Even though there is not much evidence in the literature of persistent infection as an explanation of long COVID symptoms, the authors need to explain the evidence behind the claim that nucleocapsid antibody levels can be used as a marker of persistent infection. In their own paper used as reference for this statement such an association is not shown. Please explain.

RE. Submission of revised manuscript for NCOMMS-22-49348

We thank the reviewers for their helpful comments and suggestions and here address the reviewer comments with our detailed responses:

REVIEWER COMMENTS

Reviewer #1 (Remarks to the Author):

- Many of the claims in the manuscript are quite broad and insufficient data are provided to support the expansive assertions. While there were no differences observed between the study groups analyzed in this manuscript, the immunologic assays are limited to anybody binding, neutralizing titers, and IFN γ ELISPOT analysis with 4 peptide pools spanning only a portion of the SARS-CoV-2 proteome. Fc-dependent/non-neutralizing antibody responses (such as ADCC/ADNP/ADCP activity) are not address in this analysis, and the IFN-g ELISPOT assay used in this analysis preferentially captures the cytotoxic CD8+ T cell response rather than the helper CD4+ T cell response. This is notable as SARS-CoV-2 infection and mRNA vaccination has been repeatedly shown to induce a more pronounced CD4 T cell response against Spike. These gaps do not invalidate any of the data presented, *but it is overly broad to claim that there is no difference in "immunity" when only a portion of the potential immune response is analyzed.* More precise language in the title and abstract could easily address these concerns.

Thank you for these comments, now addressed both by broadening the analysis to use of full-sequence spike megapools and by toning down the title and abstract. Please note also that the length of peptide used (20mers) are generally considered to optimally stimulate CD4 responses, though some CD8 stimulation can occur following reprocessing. In response to this comment, we opted to re-title the paper a little more specifically and less broadly: 'Persistent symptoms after COVID-19 during the first wave are not associated with differential neutralising antibody or T cell immunity to SARS-CoV-2.' We hope referee 1 will accept that it would be unwieldy to go further in listing aspects that were or were not changed or analysed, having here listed what are arguably the two measures that most would highlight.

We have also caveated the final sentence of the Abstract as follows: 'Thus, quantitative differences *in these measured parameters* of SARS-CoV-2 adaptive immunity during acute infection are unlikely to contribute to Long Covid causality'.

- Different numbers of individuals appear to have been analyzed in the different serological and cellular assays. It is unclear how these individuals were selected from the total study population and what the overlap is between the different assays.

No specific selection was imposed beyond the recovered versus persistent symptom criteria as defined in the methods. Thus, Figure 1A-C describes all the recruited samples for these groups. Figure 1D-G includes all individuals recruited at this time-point for whom sera and PBMC were available. The newly added panels (added in response to reviewer 2 comments) to compare longitudinal T cell responses to complete spike megapool (Figure

1C, H, I) encompasses all the individuals for whom weekly bleed PBMC samples were available following a PCR positive SARS-CoV-2 infection. As described at line 394, Figure 2A, B describes data from all HCW recruited at 1-year follow up who had received two vaccine doses. Data from HCW shown in Figure 2C, D are representative of the whole cohort as can be seen in our newly added Supplementary Figure 2 data.

- The use of N Ab responses as a proxy measurement of an ongoing/persistent infection is unconventional. The paper cited to support this claim is a study on SARS-CoV-2 vaccine responses and does not mention antigen persistence. The studies that have posited that persistent SARS-CoV-2 antigen influences that stability and maturation of the long-term anti-SARS-CoV-2 immune response have primarily focused on the stability and accumulation of somatic hypermutations in memory B cells.

With respect, we would suggest that one of many points holding back any agreement on persistent viral infection is lack of consensus on how to identify it. Our argument was really a ‘from 1st principles’ argument that surely, a persistent viral reservoir would necessarily be visible through its immunogenicity and impact on ongoing stimulation of the Ab response to N, and thus a tendency to increased levels. In this regard our argument was very much by analogy to the principle of an immune stimulating reservoir as seen in ref 9. This is now justified in more detail at line 133.

Reviewer #2

- This study attempts to determine if persistent symptoms after SARS-CoV-2 infection (Long-Covid) are associated with differential SARS-CoV-2 antibody or T cell responses by longitudinal analysis of healthcare workers with and without persistent symptoms. The study has some strengths. The Authors have access to a large cohort through the BARTS COVIDsortium with serial samples and weekly SARS-CoV-2 RT-PCR testing. Additionally, subjects provided a symptom diary initiated before anyone had knowledge of Long Covid, thus providing a potentially unbiased account of symptoms and retrospective grouping of subjects.

We thank reviewer 2 for pointing out a major and potentially unique strength of our study – access to this large HCW cohort with weekly blood sampling since the start of the first wave including a symptom diary that was *“initiated before anyone had knowledge of Long Covid, thus providing a potentially unbiased account of symptoms and retrospective grouping of subjects.”*

Major Concerns:

- The antibody data presented here is consistent with other published studies, but the T cell data is not. There are concerns regarding the methodology used to measure T cell responses to SARS-CoV-2. The peptide pools used were only a fraction of the full proteins examined. For example, only 18 peptides (13-20mers) were used for the Spike protein. Most studies use 170+ overlapping 15mers peptides which encompass the whole Spike protein for T cell assays. It is unclear how these peptides were chosen and how they compare to stimulations with the full set of overlapping peptides. While it is noted on page 17 this was previously described no reference is provided. The same concern about the use of a limited number of peptide holds for

Nucleocapsid, Membrane and ORF3a/7a proteins. Comparison of the limited pool of peptide used in this study to full set of overlapping peptides should be shown. The use of a limited set of peptides may result in incomplete evaluation of the actual T cell responses. The T cell responses also seem quite low compared to what is published by other groups. This is likely because the peptide pools are so small which brings up concerns about the accuracy of T cell assay. A longitudinal analysis of SARS-CoV-2-specific T cells, as was done for antibodies, would be useful and would shed light on the kinetics of the T cell response in the two groups.

We are less certain than Referee 2 that there is clear agreement from previously published studies as to whether T cell responses to SARS-CoV-2 are changed or not in those with persistent symptoms. The peptide pool used in the initial submission uses a focused pool of key epitopes that largely reiterate the behaviour of larger pools, as has been documented in a series of keynote papers by our group and others. However, noting the comments of Referee 2, we have now added new data using the full overlapping peptide set of megapools, as suggested as well as giving the citations for the validity and relevance of the mapped epitope pools (MEP, references 19-21, 24, 25). These assays support our case, similarly showing no differences. As we now add at line 162:

‘We then explored longitudinal T cell responses in HCW with PCR-confirmed infection, who subsequently did or did not report persistent symptoms, and for whom weekly PBMC samples were available. In order to be sure that we had not missed T cell response differences related to selected epitopes within the viral proteome, these studies utilised full SARS-CoV-2 epitope megapools encompassing either spike or non-spike peptides (Figure 1 H,I; Supplementary Figure 2)¹⁸. Again, no differential pattern of response between HCW with or without persistent symptoms could be seen.’

- Key references which demonstrate T cell responses are maintained in individuals with Long Covid are not included and should be added and discussed (File JK et al. - JCI Insights and Littlefield K et al. - PLoS Pathogens). These studies show that SARS-CoV-2 specific T cells are maintained and significantly elevated compared to individuals who cleared the virus and had no persistent symptoms 4 weeks post infection, contradicting the data presented in this study.

These papers and other related studies are now discussed at line 80 considering citations 11-18. In terms of the current lack of consensus and, therefore, the need for more data, we note that reviewer 3 conversely draws our attention to studies showing *reduced* T cell responses in Long COVID. We note that the cohorts in the additional papers that we now cite had included more severe Long Covid cases, in many cases following hospitalised infections, by contrast to the asymptomatic and mild infections reported by us. As we now write

‘Several studies have looked at T cell subset phenotypes and at T cell immunity to SARS-CoV-2 comparing individuals with or without Long Covid finding a number of potential differences though, as yet, no consensus¹¹⁻¹⁸. Some find evidence of enhanced SARS-CoV-2 adaptive immunity in those progressing to Long Covid: for example, among ongoing pulmonary Long Covid cases, substantially increased CD4 and CD8 responses were found¹², while another study showed a more sustained T cell and Ab response, albeit in a more severe cohort, many of whom had been hospitalised¹³. Increased convalescent antibody titres have been reported by some as a marker of Long Covid. Other cohort studies either

found no difference between groups in SARS-CoV-2 immunity¹⁵, or that reduced or rapidly declining responses were found in Long Covid¹⁶⁻¹⁸.

- It would have been useful to follow the subjects out longer in Figure 1. The antibody responses have not begun to wain in either cohort as shown in Figure 1 A+B. Figure 2 C+D does provide some useful T cell data on a very small subset of subjects (only 7 subject with persistent symptoms) at 12 months post infection and those with persistent symptoms have higher neutralization and higher number of SAR-CoV-2 specific T cells, although not statistically significant (please show the P value in the figure). Since the 12 month time point maybe the most relevant a larger cohort of subjects with persistent symptoms is needed.

We agree that it would have been valuable to have followed more of the HCW for a longer period, though within the limits of this dataset, the statement that there is no *differential* waning holds. The reality was that this was a cohort of frontline HCW at work during the first wave and by the end of the first year, there was some attrition in coming forward to give repeat blood samples. The p-value has been added as requested.

-Long haul symptoms are very heterogenous and can be broken down into 3 main categories - cardiovascular, pulmonary and neurological. This paper lumps them all together but other studies show that the various persistent symptoms may have different causes. The cohort is likely too small to subdivide by symptom type, but this could be a useful analysis, at least for data in Figure 1 where the number of subjects is fairly large. This should also be discussed.

Since this study was established on the basis of early self-reporting of symptoms prior to knowledge of Long Covid stratification we lack access to the granularity that would be required. The categories accessible to this study are shown in Table 1, with 'shortness of breath' the most common persistent symptom. The questionnaire was expanded at 12 months to include additional symptoms, with shortness of breath still the most common symptom. In order to consider this reviewer point, individuals specifically suffering from shortness of breath are indicated in purple in Figure 2.

Minor concerns:

-Has the group compared those with respiratory symptoms to those without in the persistent symptom group? Based on both tables about half of those with persistent symptoms had respiratory complications.

Yes, as discussed above and now shown in Figure 2.

- Figure 1A and B show peak Ab titer. Was the titer at week 1 or 2 examined? Speed of response would be interesting with this dataset.

This is shown in Figure 1C, where it can be seen that the speed of Ab response was not different between groups.

- Figure 2 – was there tracking to ensure there were no other illnesses or subsequent SARS-CoV-2 infection for those who “Recovered”

Yes. HCW were followed with repeat N Ab serology at 6 and 12 months; individuals who had experienced an additional SARS-CoV-2 infection between the 6 and 12 month timepoint are shown in green. They were also asked to report any PCR-positive infection.

- It states on page 4 that an advantage of this study is the capture of PCR negative controls, but none of that data is shown. How do these data compare to PCR-responses?

'Since data from our HCW negative controls have been reported in citation 16, comparison with that group was not a focus here, so that line 105 has been amended as follows: 'HCW gave longitudinal blood samples allowing us to compare immune parameters in HCW with mild or asymptomatic laboratory confirmed SARS-CoV-2 infection during the first wave.' The advantage of our screening approach is that we were able to capture and document asymptomatic infections at a time when PCR testing was still limited and LFT did not exist.

How were the cutoff for positivity determined for both antibody and T cell responses?

No specific cutoff definition was applied to T cell responses. In Figure 1 D-F we specify IC50 values for Ab and SFC ranges for T cells categorised as 'none', 'low' or 'high'. With respect to Ab responses, SARS-CoV-2 Ab testing was carried out at Public Health England UK (now UKHSA), we write at line 80 in the Supplementary Methods, 'Anti-N results are expressed as a cutoff index (COI) value based on the electrochemiluminescence signal of a two-point calibration, with results COI ≥ 1.0 classified as positive. Anti-spike results are expressed as units per ml (U/ml) similarly based on a two-point calibration and a reagent specific master curve, with a quantitative range of 0.4 to 2,500 U/ml. Samples with a value of ≥ 1.0 U/ml are interpreted as positive for spike antibodies, and samples exceeding >250 U/ml are automatically diluted by the analyzer.'

-How were the samples used in Figure 2 C and D chose from the larger cohort? Are they representative of the larger cohort? Data from Figure 1 should be used to show they are representative.

Yes, they are representative of the larger cohort. Please see our response on this point to reviewer 1 where we show in our newly added Supplementary Figure 2 that the Figure 2C and Figure 2D samples are representative of the larger cohort.

-There is discussion of EBV reactivation as an underlying factor for Long Covid. The authors say they assayed EBV specific T cell responses and found no difference. They can't say this because the CEF peptide pool is a mixture of CMV, EBV and Flu peptide and they are unable to determine which peptides are eliciting the response.

We agree that the CEF mixture offers rather tangential evidence on this point. We now write at line 160: 'No difference was seen between the persistent and recovery groups, although with the caveat that these responses also encompass CMV and flu epitopes (Figure 1G)'.

-Why wasn't a specific incubation time used for the T cell assay. It states they were incubated for 18-22 hours. It should have been the same duration for all assays.

Assays were indeed stopped at the same timepoint of 20h. The wording (18-20h) had been taken from a generic lab protocol and has now been corrected at line 142 of the Supplementary Methods.

-Showing the proportion of the response in Figure 1 and 2 that were high, low or none seems arbitrary. How were these cutoffs determined? These are not particularly helpful and could be removed.

The cutoffs are described above. While noting that such doughnut representations are not to everyone's taste, some referees and editors find the visual representation of differences helpful

-While subjects provide a self-assessment of symptoms was there any clinical diagnosis of Long Haul disease done at 12 months post infection?

No, these were HCW and clinical referral was not part of the study protocol.

-Does the group have any data on masking? Assumed high adherence in HCW population.

There was indeed high adherence of mask wearing while working in the clinical environment in the HCW population. There was no difference in PPE usage while at work comparing the individuals who subsequently fully recovered (84%) and those that developed persistent symptoms (88%).

	Used PPE at work		
	YES	NO	% YES
Recovered	51	10	84
Persistent symptoms	22	3	88
Entire covidsortium	584	145	80

Reviewer #3 (Remarks to the Author):

The manuscript is a follow-up study of a very interesting cohort of health care workers in the UK who have been followed since before SARS-CoV-2 infection in March 2022, whereby COVID symptoms could be followed from before infection and longitudinally. The authors have used standard immunological assays to confirm infection, and at follow-up 4 months post infection, and later up to 12 months. In this paper an effort has been made to categorize the HCWs in long COVID participants, fully recovered, and uninfected participants. The main conclusion is that no immunological markers linked to long COVID symptoms were found, and they conclude that differences in adaptive immunity are unlikely contributors of long COVID.

- Some questions need to be addressed:
The authors claim that the hypothesis of an immunopathogenesis is largely driven by patients, while published papers supporting such a hypothesis are not referenced at all, except a paper where the authors are themselves involved, identifying a proteome profile differing between long haulers and recovered, in the same patient cohort. The literature on differential immune responses in long COVID patients and

recovered needs to be referenced. Long term immunological differences between long COVID and recovered can be found in Peluso MJ et al Cell Rep 2021 and in Phetsouphanh C et al Nat Immunol 2022.

We apologise for the brevity in these citations which was to fit with the brief manuscript format. In response to this comment as well as other reviewers, we now include some lines of 'mini-review' on the rather divergent literature on immunity to SARS-CoV-2 in Long Covid at line 80:

'Several studies have looked at T cell subset phenotypes and at T cell immunity to SARS-CoV-2 comparing individuals with or without Long Covid finding a number of potential differences though, as yet, no consensus¹¹⁻¹⁸. Some find evidence of enhanced SARS-CoV-2 adaptive immunity in those progressing to Long Covid: for example, among ongoing pulmonary Long Covid cases, substantially increased CD4 and CD8 responses were found¹², while another study showed a more sustained T cell and Ab response, albeit in a more severe cohort, many of whom had been hospitalised¹³. Increased convalescent antibody titres have been reported by some as a marker of Long Covid. Other cohort studies either found no difference between groups in SARS-CoV-2 immunity¹⁵, or that reduced or rapidly declining responses were found in Long Covid¹⁶⁻¹⁸.'

2. Literature showing an association between low acute immune responses (Garcia-Abellan J et al J Clin Immunol 2021, Lerum TV et al Sci Rep 2021) or high peak convalescent antibodies (Blomberg B et al Nat Med 2021) are interesting in relation to the present study, and raises the question of whether later time point such as the 4 month time point, with potential waning immune responses, used in this study, can explain differences in findings. Note also Cervia C et al Nature Comm 2022. This needs to be discussed.

These findings are discussed from line 84, though it is by no means clear that differential findings in the previous studies relate to choice of timepoints.

3. The main weakness of the study is the low sample size. With only 25 long COVID patients, where even one of the most persistent symptoms which is cognitive symptoms, such as memory and concentration problems, are not addressed, makes firm conclusions hard to make, and the conclusions should be modified accordingly.

We agree and we have moderated our comments accordingly. It is both a strength and weakness of this study that the protocol for recording symptoms pre-dates the current consensus symptom list.

4. Even though there is not much evidence in the literature of persistent infection as an explanation of long COVID symptoms, the authors need to explain the evidence behind the claim that nucleocapsid antibody levels can be used as a marker of persistent infection. In their own paper used as reference for this statement such an association is not shown. Please explain.

We now attempt to argue this point more clearly at line 130, as follows:

'it also allowed us to use the trajectory of the longitudinal N Ab response as a proxy measurement of whether there was likely to be an ongoing, persistent reservoir of virus⁹⁻¹¹. Viral persistence would be predicted to correlate with a sustained or rising N Ab response; from first principles, a persistent viral reservoir would necessarily be visible through its

immunogenicity and impact on ongoing stimulation of the Ab response to N, and thus a tendency to increased levels.'

REVIEWERS' COMMENTS

Reviewer #2 (Remarks to the Author):

Use of mega pools address my main concerns.